# Monomeric ephrinB2 binding induces allosteric changes in Nipah virus G that precede its full activation

Joyce J.W. Wong[1], Tracy A. Young[2], Jiayan Zhang[3,4,5], Shiheng Liu[4,5], George P. Leser[6,7], Elizabeth A. Komives[8], Robert A. Lamb[6,7], Z. Hong Zhou[4,5], Joshua Salafsky[2] & Theodore S. Jardetzky[1]

Nipah virus is an emergent paramyxovirus that causes deadly encephalitis and respiratory infections in humans. Two glycoproteins coordinate the infection of host cells, an attachment protein (G), which binds to cell surface receptors, and a fusion (F) protein, which carries out the process of virus-cell membrane fusion. The G protein binds to ephrin B2/3 receptors, inducing G conformational changes that trigger F protein refolding. Using an optical approach based on second harmonic generation, we show that monomeric and dimeric receptors activate distinct conformational changes in G. The monomeric receptor-induced changes are not detected by conformation-sensitive monoclonal antibodies or through electron microscopy analysis of G:ephrinB2 complexes. However, hydrogen/deuterium exchange experiments confirm the second harmonic generation observations and reveal allosteric changes in the G receptor binding and F-activating stalk domains, providing insights into the pathway of receptor-activated virus entry.

[1] Department of Structural Biology, Stanford University School of Medicine, Stanford, CA 94305, USA. [2] Biodesy, Inc., South San Francisco, CA 94080, USA. [3] Molecular Biology Institute, University of California Los Angeles, Los Angeles, CA 90095, USA. [4] Department of Microbiology, Immunology & Molecular Genetics, University of California Los Angeles, Los Angeles, CA 90095, USA. [5] California NanoSystems Institute, University of California Los Angeles, Los Angeles, CA 90095, USA. [6] Howard Hughes Medical Institute, Northwestern University, Evanston, IL 60208-3500, USA. [7] Department of Molecular Biosciences, Northwestern University, Evanston, IL 60208-3500, USA. [8] Department of Chemistry and Biochemistry, University of California San Diego, San Diego, CA 92093, USA. Correspondence and requests for materials should be addressed to T.S.J. (email: tjardetz@stanford.edu)

The *Paramyxoviridae* are enveloped, negative-strand RNA viruses that infect both humans and animals. The family includes many important human pathogens such as mumps virus, measles virus, respiratory syncytial virus, human metapneumovirus, parainfluenza viruses, Nipah virus, and Hendra virus[1]. Nipah and Hendra viruses, which define the Henipavirus genus within the *Paramyxoviridae*, cause periodic outbreaks of encephalitic and respiratory illness in humans with high morbidity and mortality. They are therefore designated BSL-4 agents and no human vaccines or therapeutics are currently available for these viruses.

The entry of paramyxoviruses into host cells requires the merger of the viral lipid envelope and the host cell plasma membrane[1, 2]. For the Henipaviruses, this process is triggered by specific binding of the attachment protein, G, to its host cell receptors, ephrinB2 or ephrinB3[3–5]. The G protein is a tetramer with an N-terminal transmembrane domain, an extramembrane stalk domain, consisting of a membrane proximal alpha-helical portion and distal proline-rich portion[6], and a C-terminal globular receptor-binding domain (RBD) that has a 6-bladed beta-propeller fold[7, 8]. EphrinB2/B3 are members of the B-class of ephrin ligands, which endogenously function in cell signaling via interaction with EphR receptors. B-class ephrins are transmembrane proteins with a receptor-binding ectodomain that are thought to form loose dimers[9]. The current prevailing model for Henipavirus fusion is that G forms complexes with the fusion (F) protein prior to fusion triggering[10–12]. Binding of ephrin B2/3 is thought to result in a series of rearrangements in the G tetramer that culminate in the exposure of stalk domain residues that activate F, but are occluded by the head domains in the untriggered state[11, 13, 14]. These rearrangements are thought to involve changes in the quaternary structure of the G tetramer, as very little change occurs within the henipavirus G RBDs in crystal structures with ephrinB2 or B3[15, 16]. Contacts between exposed G stalk residues and F are hypothesized to then cause F to transition from its pre- to postfusion state, drawing the host and viral membranes together to promote bilayer fusion.

Evidence for receptor-induced Nipah virus G (NiV G) rearrangements comes from changes in NiV G circular dichroism, Raman spectra, and binding affinity to a panel of conformation-specific anti-NiV G antibodies[13, 14, 17–19]. Notably, an epitope immediately preceding the N-terminus of the NiV G RBD recognized by monoclonal antibody mAb45 and a stalk domain epitope recognized by polyclonal antibody Ab167 become more tightly bound by their respective antibodies, while an epitope on the globular head domain recognized by monoclonal antibody mAb213 becomes less bound[13, 17]. Similarly, a panel of monoclonal antibodies specific to Hendra G (HeV G) shows increases in binding to HeV G upon ephrinB2 addition[20]. Site-directed mutants throughout the NiV G ectodomain exhibit different profiles for binding to these conformational antibodies. These data, combined with the mutant profiles for fusion activity, receptor binding, and F binding, has led to a model in which a multi-step cascade of G conformational changes occurs during fusion activation, which may be arrested at various stages by the G mutants[13, 14]. However, little is known about the details of these conformational changes and how ephrin B2/3 binding to the G RBDs leads to their induction.

Receptor-induced conformational changes in the attachment proteins of other members of the paramyxovirus family are also thought to be part of their fusion mechanisms. The crystal structures of the PIV5 and NDV HN ectodomains and the measles virus H receptor binding domains (RBDs) exhibit varying orientations with respect to each other[21–23]. In addition, restraining the H protein stalk helices of measles virus H and

canine distemper virus H with inter-chain disulfide bonds disrupts fusion[24, 25]. The F-activating role of the exposed attachment protein stalk is supported by constitutive activation of paramyxovirus fusion with headless attachment protein constructs[13, 26–28], fusion activation by chimeric attachment proteins with heterotypic head domains and homotypic stalks[29] and the disruption of fusion by stalk domain mutants that modulate interactions with F[30–32].

To understand better the conformational changes induced in NiV G upon binding its host receptor ephrinB2, we explored the possibility of using second harmonic generation (SHG) to monitor G conformational states. Since the concept of using SHG to study proteins at an interface was introduced[33], it has been used to detect a variety of ligand-induced conformational changes in proteins that are tethered to a planar surface[34–37]. Proteins are labeled with SHG-active dye molecules, which are covalently attached to protein side chains and provide a reporter signal that is sensitive to the dye orientation and conformational changes[33]. The SHG signal is very sensitive to small changes in SHG label orientation, being proportional to $<\cos^3\theta>^2$, where $\theta$ is the angle between the SHG label's molecular dipole and the surface normal and the brackets denote an orientational average over all molecules under illumination by the laser beam. The SHG signal is thus sensitive to both the mean angle and the width of the dye conformational distribution and is therefore potentially capable of detecting changes in protein dynamics in the absence of net conformational changes[34]. Subtle changes in this distribution due to changes in conformation result in changes in SHG intensity with a good signal to noise ratio. Therefore, distinguishing NiV G states in terms of both net conformation and dynamic flexibility is potentially possible using this technique. Previous reports have shown that non-specific lysine labeling at multiple sites still allows ligand-induced changes to be detected readily[34, 35, 38].

Here we study the ephrinB2-induced conformational changes in NiV G using SHG, hydrogen-deuterium exchange mass spectrometry (HDX-MS), antibody binding, and electron microscopy (EM). SHG studies indicate that monomeric ephrinB2 (comprising only the globular portion of its ectodomain) induces conformational changes in G that are distinct from dimeric ephrinB2 (a chimera with an Fc domain that mediate dimerization). Surprisingly, monomeric ephrinB2 does not significantly alter the binding of two anti-G conformational antibodies as compared to ephrinB2-Fc. Consistent with the mAb binding results, EM analysis of NiV G complexed with monomeric ephrinB2 also revealed no large-scale conformational changes. Because the SHG data indicated that G undergoes conformational changes upon monomeric ephrinB2 binding that are not detected by conformation-sensitive antibodies, we used HDX-MS to study the impact of receptor binding on G. HDX-MS analysis of G reveals significant differences in H/D exchange for peptides in the globular RBD and in the N-terminal stalk domain upon binding to monomeric ephrinB2, indicating that allosteric changes are induced, consistent with the SHG observations. Together, these data support a model in which monomeric receptor binding induces allosteric changes in individual G subunits, conveyed through the RBD to the helical stalk domains, which prime the tetramer to undergo further conformational changes induced by oligomeric receptors to promote F-mediated membrane fusion. We conclude that SHG provides a powerful and sensitive approach to detecting subtle conformational and dynamic changes in protein structures that complements other structural and biophysical techniques. In addition, our EM analysis of the NiV G tetramer reveals a different arrangement of RBDs as compared to crystal structures of the ectodomains of HN proteins from Newcastle disease virus and parainfluenza virus 5.

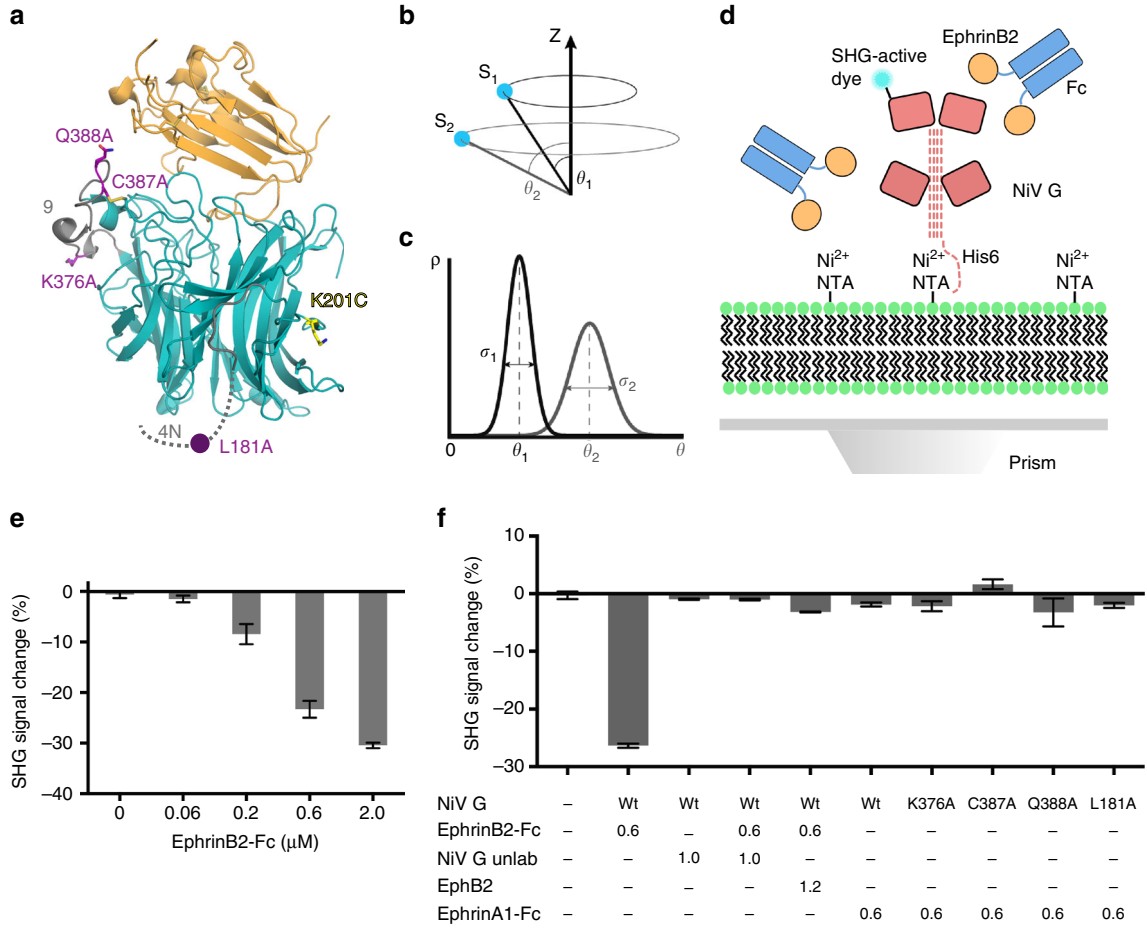

**Fig. 1** Detection of ephrinB2 binding to NiV G by SHG. **a** Crystal structure of the globular domain of NiV G bound to the globular domain of ephrinB2. NiV G conformational mutants are indicated in *purple*, and the dominant labeling site of NiV G is indicated in *yellow*. Epitopes for conformation-sensitive antibodies mAb45 and mAb213 are indicated in *gray*. **b** Prior to ligand binding, the SHG label representing protein conformational ensemble S1 has an average orientation angle $\theta_1$ relative to the surface normal. Ligand binding elicits protein conformation ensemble S2 with an average label orientation $\theta_2$. **c** The SHG signal depends on both the average label orientation $\theta$ and the width of the orientational distribution $\sigma$ of the probe. Rearrangements in protein structure upon ligand binding that result in changes of either $\theta$, $\sigma$, or both will be reflected in the SHG signal change. **d** Biodesy read plate setup. A NiV G ectodomain construct with an N-terminal His6-tag was oriented in a fixed manner via binding of the His6-tag to Ni-NTA groups on a supported lipid bilayer. The globular domains are represented by *pink rectangles*, and the stalk domain by *dashed pink lines*. **e** Δ SHG concentration-dependent dose-response of ephrinB2-Fc binding to NiV G. His6-tagged NiV G ectodomain construct was bound to Ni-NTA-containing supported lipid bilayer at 0.5 μM. EphrinB2-Fc was added to the indicated final concentrations. **f** Negative controls for SHG response in NiV G ectodomain constructs. NiV G was bound at 0.5 μM and ephrinB2 constructs and competitors were added at indicated amounts (μM). Mean and s.d. values for SHG data are shown from a representative experiment, $n = 3$

## Results

**SHG signal change in NiV G in response to ephrinB2 binding.** Wildtype NiV G ectodomain was labeled through free amino groups with SHG-active dye in order to provide a reporter for potential conformational changes. The percentage of total labeling at each detected lysine residue showed modification of a dominant site in the NiV G head domain, K201 (Fig. 1a and Supplementary Table 1). Production of an SHG signal depends on the non-isotropic orientation of the SHG-active label with respect to a surface or interface (Fig. 1b), as well as the width of the orientational distribution (Fig. 1c). We used the Biodesy Delta instrument for measuring SHG, where NiV G ectodomains were captured through N-terminal His6 tags by Ni-NTA groups in a supported lipid bilayer (Fig. 1d).

The specificity of the NiV G SHG response to ephrinB2 was tested in a dose-response titration of ephrinB2-Fc, competition experiments, and with a non-NiV G reactive ephrin isotype, ephrinA1. Wildtype NiV G showed a greater negative SHG response (ΔSHG) to increasing concentrations of ephrinB2-Fc

(Fig. 1e and Supplementary Fig. 1). Pre-incubation of ephrinB2-Fc with known competitors of NiV G for ephrinB2 binding (unlabeled wildtype NiV G or the ephrinB2 receptor, EphB2) greatly reduced the observed ΔSHG, consistent with their competition for ephrinB2 binding to NiV G (Fig. 1f).

**NiV G mutants exhibit distinct SHG signal changes.** Four site-directed mutants of NiV G with differing conformational antibody binding profiles were chosen for analysis by SHG relative to wildtype: K376A, C387A, Q388A, and L181A (Fig. 1a). K376A, C387A, and Q388A are part of region 9 (residues 371–392), which shows decreased binding to mAb213 upon ephrinB2 binding to NiV G[13]. L181A is near region 4N (195–211), which shows increased binding to mAb45 in the presence of ephrinB2. Region 9 is believed to undergo an ephrinB2-triggered conformational change before region 4N. Prior studies of antibody binding to the G mutants in the absence of receptor indicated wildtype behavior for K376A, lower overall

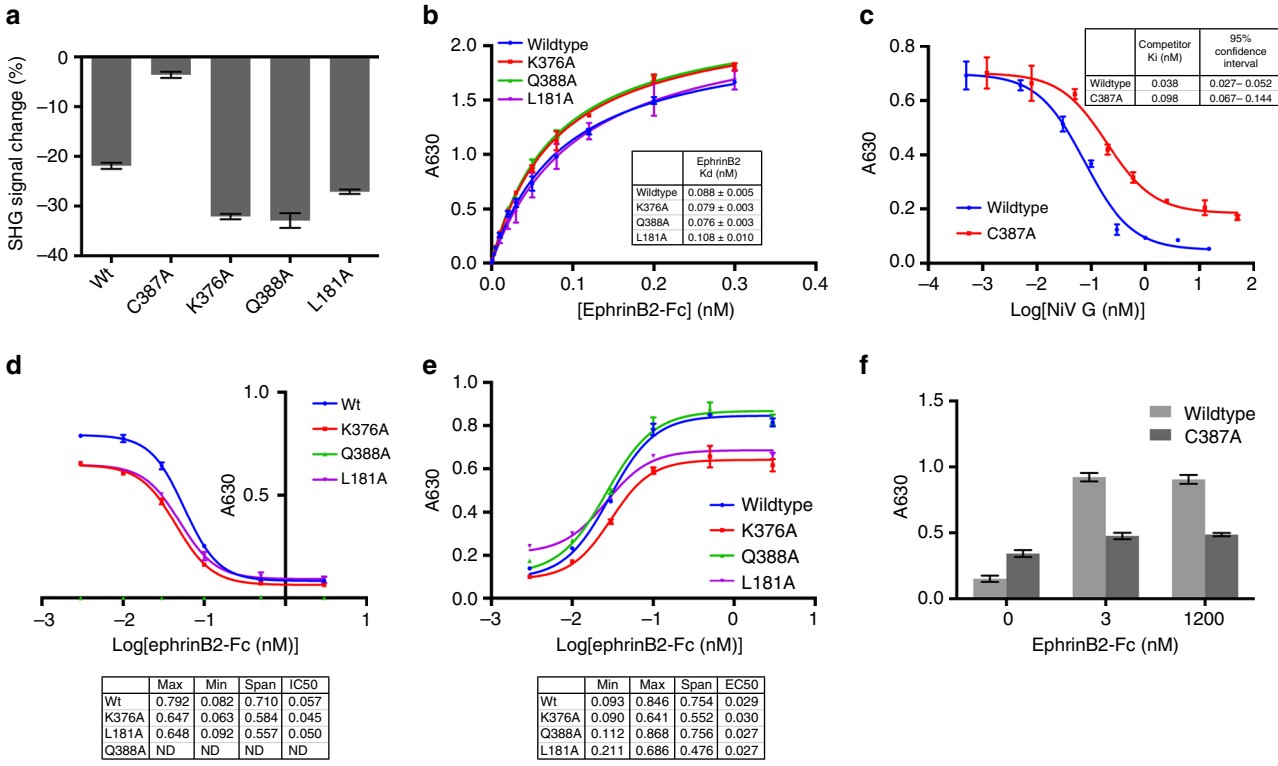

**Fig. 2** Response of NiV G mutants to ephrinB2-Fc binding measured by SHG and conformational antibody binding. **a** Change in SHG signal of NiVG ectodomain constructs bound at 0.5 μM at 20 min following ephrinB2-Fc addition to 0.6 μM. **b** Binding affinity of ephrinB2-Fc to NiV G ectodomain constructs measured by direct ELISA. **c** Inhibition of wildtype NiV G binding to ephrinB2-Fc by wildtype NiV G and a C387A mutant. **d** Binding of mAb213 to NiV G constructs with increasing ephrinB2-Fc concentration. **e** Binding of mAb45 to NiV G constructs with increasing ephrinB2-Fc concentration. **f** Binding of mAb45 to wildtype NiV G and a C387A mutant with increasing ephrinB2-Fc concentration. Mean and s.d. values for SHG data are shown from a representative experiment, $n = 3$. Mean and s.d. values for ELISA data are shown from a representative experiment, $n = 2$

binding for C387A to all antibodies, a large decrease in binding to mAb213 for Q388A, and an increase in binding to mAb45 for L181A. In the presence of ephrinB2, C387A showed relatively little change in mAb213 and mAb45 binding compared to wildtype. L181A showed a similar decrease in mAb213 binding to wildtype and an unchanged level of mAb45 binding[13]. Overall, these data suggested that the conformations and receptor-induced changes in the mutants were distinct. We were therefore interested in examining these mutants using SHG to determine whether they exhibited distinct conformational signatures indicative of altered conformations.

The mutant G proteins were purified and labeled similarly to the wildtype protein, with degrees of labeling of $1.67 \pm 0.15$. Liquid chromatography tandem mass spectrometry (LC-MS) analysis of the labeled proteins indicated that the wildtype and mutant proteins were labeled mostly similarly across all constructs, though substantial differences exist for several residues, notably K130, K386, and K415 (Supplementary Table 1).

None of the mutants responded to the control ephrinA1-Fc (Fig. 1f), but three of them (K376A, Q388A, and L181A) showed comparable, but distinct, SHG signal changes in the presence of ephrinB2-Fc (Fig. 2a). The C387A mutant did not show a significant SHG response in the presence of 0.6 μM ephrinB2-Fc, but at 10 μM ephrinB2-Fc, the C387A mutant yielded ΔSHG comparable to wildtype G (−21.9%; Supplementary Fig. 1). K376A and Q388A showed similar ΔSHG values to each other (−32.9 and −31.9%, respectively), which were greater than those from the wildtype G (Fig. 2a, Supplementary Fig. 2). L181A showed a ΔSHG between wildtype and the K376A or Q388A mutants, (−27.1%). The variable ΔSHG values observed could be consistent with the mutants initiating from different

conformational states, as previously proposed, but we cannot exclude the possibility that some variability in dye labeling may contribute as well. Nonetheless, these data indicate that all of the G mutants are capable of undergoing ephrinB2-Fc induced conformational changes.

**EphrinB2 binding affinity of NiV G mutants**. The observed SHG responses for the wildtype and G mutants could potentially arise from differences in binding affinity of ephrinB2 for the mutants, in particular for C387A[38]. We measured direct binding of ephrinB2 to the wildtype and mutant proteins using an oriented enzyme-linked immunosorbent assay (ELISA) assay. All of the mutants, except for C387A, show similar ephrinB2-Fc binding affinities to wildtype, which has a $K_d$ of 0.088 nM (Fig. 2b). In contrast, binding data for the C387A mutant did not reach saturation at μM concentrations of ephrinB2-Fc, yielded inconsistent replicates, and could not be fit to standard binding curve models. Therefore, we used a competitive binding assay to measure C387A binding affinity relative to wildtype. C387A competed with wildtype NiV G with weaker $K_i$ (Fig. 2c). We therefore conclude that the lack of SHG signal response in C387A at low ephrinB2-Fc concentrations is most likely due to weaker ephrinB2 binding affinity (consistent with NiVG-ephrinB2 binding levels reported in ref. [13]) rather than differing conformational response. For all of the other tested mutants, the similarity in binding affinity indicates that the observed differences in SHG response are due to other causes.

**EphrinB2-induced changes in mAb213 and mAb45 binding to NiV G.** Previous studies have examined conformational changes

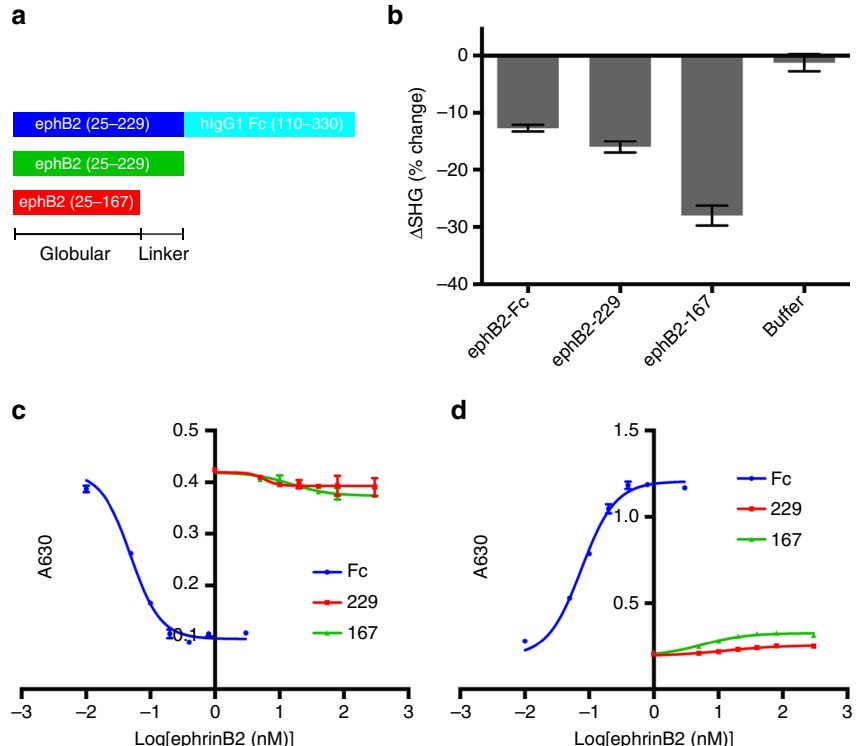

**Fig. 3** Response of wildtype NiV G to binding of ephrinB2 constructs with varying length and oligomerization measured by SHG and conformational antibody binding. **a** EphrinB2 constructs. Listed from *top* to *bottom* are ephrinB2-Fc, ephrinB2-229, and ephrinB2-167. **b** Change in SHG signal of NiVG bound at 0.5 μM at 20 min following ephrinB2-Fc addition to 1.2 μM. **c** Binding of mAb213 to wildtype NiV G with increasing ephrinB2 construct concentration. **d** Binding of mAb45 to wildtype NiV G with increasing ephrinB2 construct concentration. Mean and s.d. values for SHG data are shown from a representative experiment, $n = 3$. Mean and s.d. values for ELISA data are shown from a representative experiment, $n = 2$

in the full length, membrane embedded NiV G protein, using the conformation-sensitive antibodies mAb213 and mAb45[13]. To determine whether the soluble, NiV G ectodomain exhibits similar changes, all of the NiV G ectodomain constructs were tested for binding to mAb213 and mAb45 in the presence of increasing amounts of ephrinB2-Fc. Wildtype and two mutants (K376A and L181A) showed decreases in mAb213 binding as ephrinB2 is titrated (Fig. 2d), consistent with the previously reported studies of full-length G[13]. The Q388A mutant showed no binding to mAb213, which is consistent with the mutation destroying the antibody epitope as previously reported (Fig. 2d).

mAb45 binding increases after ephrinB2-Fc binding to full-length NiV G and we observe a similar increase with the NiV G ectodomain proteins (Fig. 2e). However, it is notable that L181A has significantly higher mAb45 initial binding prior to ephrinB2-Fc addition and saturates at a lower level compared to wildtype. This is consistent with previous observations made with full-length G. These differences were interpreted to mean that L181A is prematurely triggered, at least partially revealing the mAb45 epitope early so that the span of signal change between the receptor bound and unbound states is smaller compared to wildtype[13]. C387A binding to mAb45 is minimal at 1.2 μM ephrinB2 (Fig. 2f), consistent with its weak binding to ephrinB2-Fc and lack of SHG response. Overall these data demonstrate that ephrinB2-Fc binding to secreted wildtype and mutant G ectodomains recapitulates the antibody-sensitive conformational changes observed with the full-length G.

**Negative stain EM analysis of wildtype and mutant NiV G.** Given the differences in ΔSHG, antibody binding and receptor binding that we observed for the mutants and wildtype NiV G, we investigated whether any structural differences could be observed

by negative stain EM. Micrographs of the wildtype and mutant NiV G reveal that all of the proteins form well-defined tetramers with globular domains positioned around a central stalk (Supplementary Fig. 3). Individual particles of the tetramers suggested that the G RBDs are more separated from each other compared to the dimer-of-dimers observed in the crystal structures of its paramyxovirus homologs, PIV5 and NDV HN[22, 23]. Comparisons of the wildtype and mutant NiV G particles do not indicate any large-scale conformational differences that could account for the observed differences in antibody binding, ΔSHG or ephrinB2 binding. Notably, this includes the L181A mutant, for which mAb45 binding differences were interpreted as evidence for its adoption of a pre-triggered conformation prior to receptor binding[13] and for which similar effects were observed with our secreted ectodomain construct. The EM data indicate that the potential conformational differences detected by antibody binding may be relatively small. To visualize the effects of ephrinB2-Fc binding to G, complexes of NiV G with ephrinB2-Fc were prepared and isolated using size exclusion chromatography (Supplementary Fig. 6a) for EM studies. However, the EM samples appeared non-homogenous and aggregated (Supplementary Fig. 6b), most likely due to the ability of the dimeric ephrinB2-Fc construct to cross-link multiple NiV G tetramers.

**EphrinB2 oligomeric states induce distinct changes in NiV G.** We next asked if monomeric ephrinB2 binding to NiV G induces similar SHG and conformation-sensitive antibody binding responses. We generated two monomeric ephrinB2 constructs. EphrinB2-167 consists of only the NiVG-binding globular domain observed in crystal structures[8], residues 25–167 (Fig. 3a). The second construct, ephrinB2-229, contains the entire ephrinB2 ectodomain (residues 25–229), which are also included

in the ephrinB2-Fc fusion constructs (Fig. 3a). Strikingly, the shorter, monomeric ephrinB2-167 yielded the largest ΔSHG (−27.9%), while the longer monomeric construct (ephrinB2-229) was more similar to the dimeric receptor with ΔSHG values of −15.9 and −12.9%, respectively (Fig. 3b and Supplementary

Fig. 4). The binding affinities of ephrinB2-167 and ephrinB2-229 are 11.9 nM and 33.2 nM, respectively, within the same order of magnitude of the $K_d$ previously reported[8] (Supplementary Fig. 5), and weaker than the binding of the bivalent ephrinB2-Fc. The SHG data indicate that the monomeric ephrins induce conformational changes in G, although the magnitude of the conformational changes cannot be easily inferred from the magnitude of the ΔSHG observed. Relatively small changes in the reporter dye orientation or the distribution of protein dynamic states could yield significant ΔSHG changes, with sensitivity to 1 Å changes in structure as previously reported[37].

To determine whether the monomeric ephrinB2 proteins also induce conformational changes detected by the mAb45 and mAb213 antibodies, we conducted binding studies of the antibodies in the presence of increasing concentrations of the three receptor constructs. mAb213 binding to NiVG is unaffected by the monomeric ephrinB2 constructs even at >10× the $K_d$, while parallel experiments with the ephrinB2-Fc exhibited the expected decrease in mAb213 binding at <10× its $K_d$ (Fig. 3c). Similarly, ephrinB2-Fc induces a large increase in mAb45 binding, while the monomeric ephrinB2 proteins show little to no effects (Fig. 3d). The ephrinB2-167 construct shows slightly increased mAb45 binding compared to the longer ephrinB2-229 construct, but this is insignificant as compared to the effect of the dimeric ephrinB2-Fc. The mAb binding data suggest that monomeric ephrin cannot induce the full conformational transitions in G. However, the ΔSHG from the monomeric receptor binding indicated that conformational or dynamic changes in G are still induced, which could precede larger changes triggered by ephrinB2-Fc binding.

**Negative stain EM of apo- and ephrinB2 monomer-bound NiV G.** To directly examine potential conformational changes caused by monomeric receptor binding, NiVG complexes with ephrinB2-167 were examined by negative stain EM. The complexes were prepared by incubation with excess ephrinB2 followed by purification on a size exclusion column (Supplementary Fig. 6a). These complexes were then visualized by negative stain EM for comparison with the apo-form.

2D class averaging of unbound NiV G particles revealed that it forms an asymmetric tetramer, with a dimer of globular domains at its apical end and 2 monomeric globular domains on either side of its central stalk, with no dominant classes. There is some variability in the rotation of the upper and lower pair of dimers with respect to each other, and in the distance between the lower pair of head domains, indicating some degree of conformational flexibility in the tetramer (Fig. 4a, c).

2D class averaging of ephrinB2-167:G complexes showed a similar distribution of tetramer conformations with no obvious change from unbound NiV G (Fig. 4b, c). The bound ephrinB2-167 monomer is visible as additional density at the membrane-proximal side of the lower pair of globular domains, but no additional density can be seen at the upper 2 domains (Fig. 4c). These results further indicate that binding of monomeric ephrinB2 to NiV G is insufficient to cause large-scale conformational changes in the tetramer.

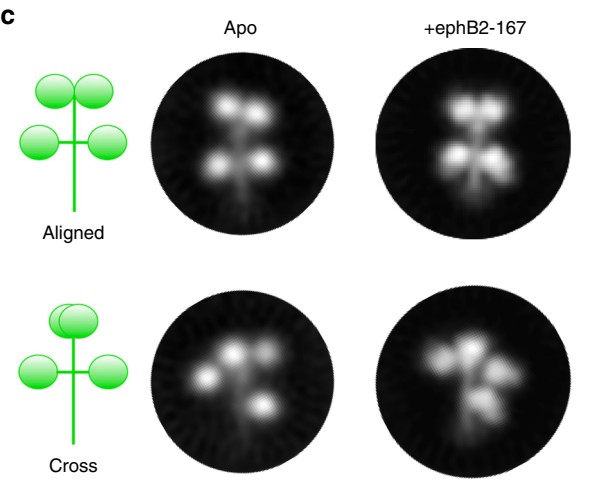

**Fig. 4** Ectodomain of NiV G visualized by negative stain electron microscopy in the presence and absence of monomeric ephrinB2. **a** 2D class averages of apo-NiV G. Class distribution proportion is indicated for each class. **b** 2D class averages of NiV G bound to ephrinB2-167. Class distribution proportion is indicated for each class. **c** Representative 2D class averages of apo- and ephrinB2-167-bound NiV G, indicating flexibility between the lower and apical pairs of NiV G globular domains

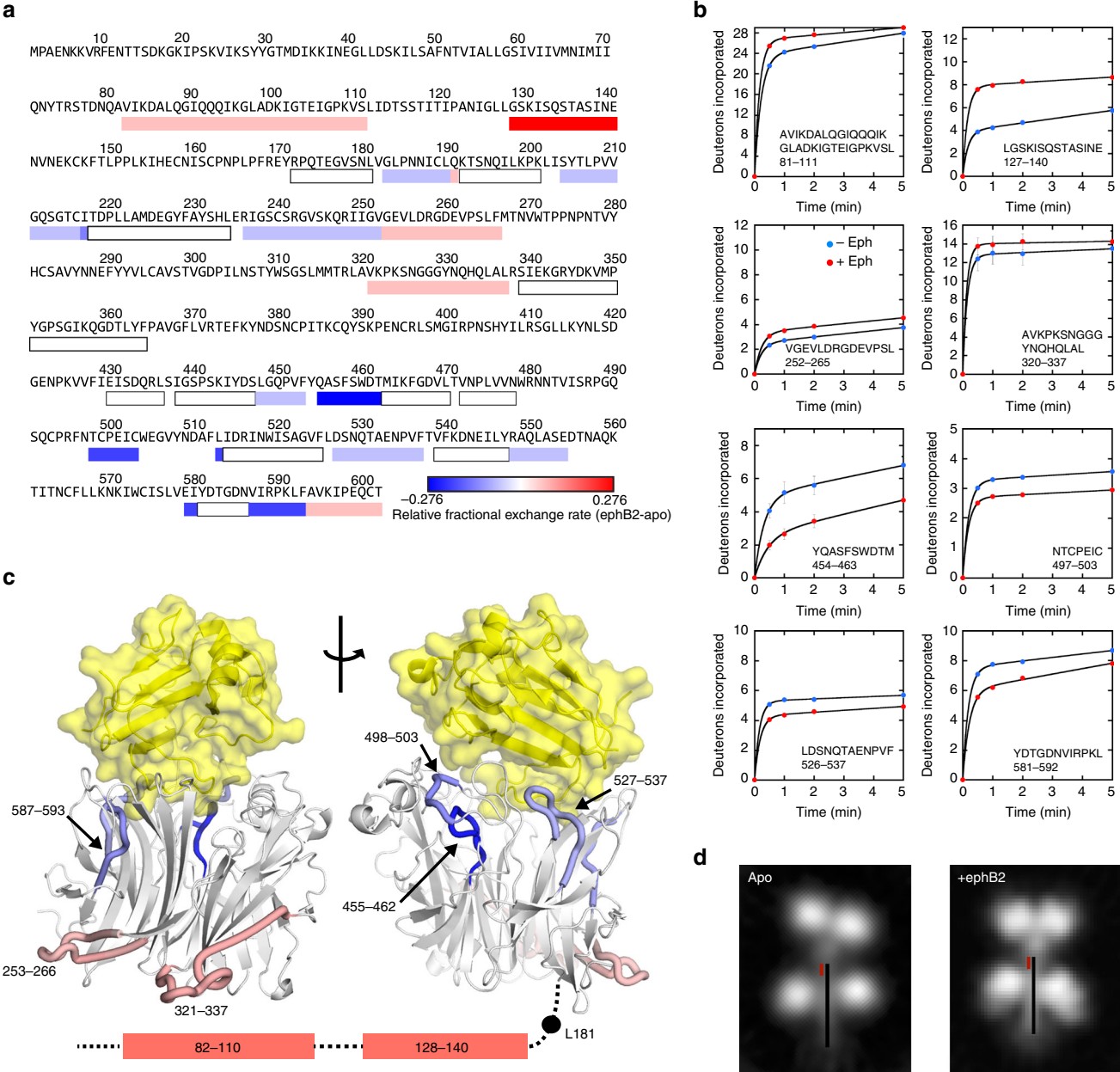

**Fig. 5** Hydrogen-deuterium exchange mass spectrometry of the NiV G ectodomain in the presence and absence of monomeric ephrinB2. **a** Relative fractional exchange rates of NiV G residues. Residue ranges were defined by the heatmap generated with DynamX. **b** Hydrogen-deuterium exchange rates for selected NiV G peptides from apo- and ephrinB2-167-bound NiV G. Mean and s.d. values for deuterium uptake are shown for each time point, $n = 3$. **c** Relative fractional exchange rates of key NiV G regions mapped onto the crystal structure of the NiV G globular domain bound to monomeric ephrinB2. Residue ranges were defined as in **a**. **d** Estimated location of the predicted α-helical stalk region of NiV G (*black line*) superposed on a representative 2D class average of NiV G obtained by electron microscopy. The estimated location of peptide 127–140, which undergoes the greatest increase in exchange upon ephrinB2-167 binding is shown with a *red line*

**HDX-MS of apo- and ephrinB2-167-bound NiV G**. To further test the possibility that monomeric ephrin binding induces changes in NiV G and to gain insight into specific regions of G that undergo changes upon ephrinB2 binding, we conducted HDX-MS experiments in the presence and absence of ephrinB2-167. The shorter ephrinB2 construct was used to minimize potential peptide overlaps with G. The H/D exchange differences between ephrinB2-bound and unbound G in peptide regions throughout G are shown in a composite heat map of peptide exchange rates in Fig. 5a. Time-dependent data on individual peptides showing notable differences in exchange rate are shown in Fig. 5b.

The peptide exchange data reveal a wide variety of exchange differences in both the head and stalk domains due to monomeric ephrinB2 binding. Some peptides are unaffected by the presence of receptor (white). Other peptides show reduced exchange when receptor is bound (blue), indicating potential stabilization of the G structure, while other peptides show significant increased exchange in the presence of receptor (red). Strikingly, the predicted alpha-helical region of the stalk contains two peptides that define one of the predominant regions with increased H/D exchange (Fig. 5a). These peptides span residues 81–111 and 127–140. These residues are not involved in ephrinB2 binding, indicating that receptor binding induces an allosteric change in

the G RBD that is propagated to the stalk. Peptides spanning residues 255–266 and 321–337, which lie within the RBD, also show increased H/D exchange in the presence of receptor (Fig. 5a–c). These residues form a contiguous region on the ephrinB2-distal face of the beta-propeller domain, which could be involved in transmitting the allosteric change to the G stalk domain. Finally, RBD loops that are in close proximity to the ephrinB2-binding interface show the most overall decrease in exchange (Fig. 5c), consistent with the direct stabilization of these regions by ephrinB2 binding. Peptides from the receptor binding loops with some of the largest decreases in H/D exchange correspond to residues 454–463, 497–503, and 526–537 (Fig. 5b, c). The largely beta-strand region of residues 588–593 shows the next largest decrease in solvent exchange with ephrinB2 binding. The peptide that undergoes by far the largest increase in solvent exchange, 127–140, maps to a potentially solvent-exposed region between pairs of RBDs in both the apo- and ephrinB2-bound forms (Fig. 5d). The H/D exchange data indicate that monomeric ephrinB2 binding induces both local and distributed changes in the G structure, consistent with the SHG observations.

## Discussion

The prevailing model for the activation of Henipavirus entry is that sequential conformational changes in G occur in response to ephrinB2 binding, forming a pathway to F protein activation. Previous observations support a model in which these changes in the G protein conformation culminate with the head domains of G moving to free its stalk domain to interact with F in a manner that activates its refolding[13, 14]. The structural nature of the conformational changes and the residues of G involved in leading to this activated state have been unclear. To address these gaps, we have employed multiple techniques to probe for conformational changes in NiV G in response to ephrinB2 binding: SHG, antibody reactivity profiling, HDX-MS, and EM.

We observe that induced conformational changes in the NiV G ectodomain are dependent on the ephrin oligomerization state. Both monomeric and dimeric ephrinB2 trigger G conformational changes measured by SHG, with the monomeric receptor yielding an overall larger signal. However, monomeric ephrinB2 binding to G fails to induce conformational changes detected by two conformation-sensitive mAbs (mAb45 and mAb213). In contrast, dimeric ephrinB2-Fc induces changes in mAb45 and mAb213 binding to secreted G ectodomain similar to those in previously studied full-length NiV G. We also observed differences in ΔSHG and mAb binding with G mutants that have been suggested to affect the initial G conformational state. However, negative stain EM studies of the wildtype and mutant G proteins suggest that these potential conformational differences may be small relative to the overall G architecture. Negative stain EM of monomeric receptor-bound G also failed to identify any significant changes in the G structure relative to uncomplexed G, consistent with the absence of conformational changes monitored by mAb binding. Although monomeric receptor binding does not induce significant changes in mAb binding, the large ΔSHG indicated that G still undergoes a significant change in conformation or dynamics.

To further probe the effects of monomeric ephrinB2 on NiV G, we used HDX-MS to examine changes in the H/D exchange profiles of G-derived peptides. Monomeric ephrinB2 binding induced increases and decreases in H/D exchange in peptides distributed throughout the G sequence and structure, pointing to allosteric effects on peptides distant from the receptor binding site. The largest increase in exchange was in a stalk region containing residues 128–140, and the largest decrease in exchange was an ephrinB2-protected loop within its binding pocket,

455–462. Two other regions that exhibit increases in exchange map to adjacent loop regions (residues 253–266 and 321–337) on the same the side of the G RBD (Fig. 5c) and these are distant from the receptor binding site, indicating that receptor binding induces allosteric changes within the beta-propeller domain.

The SHG, mAb binding, EM, and HDX-MS data support a model in which ephrin-B2 monomer binding to G induces allosteric changes that propagate from the receptor-binding site to the stalk domain, without inducing the full conformational change caused by oligomeric receptor. The increased exchange of RBD residues 253–266 and 321–337 may identify a region of the RBD that interacts directly with the stalk domain, thereby influencing the stability and H/D exchange of stalk peptides 81–111 and 127–140. Monomeric receptor binding appears to induce changes in individual G subunits that would prime further conformational changes induced by oligomeric receptor, potentially reducing the strength of RBD:stalk interactions to allow subsequent conformational rearrangements.

Since the net conformation of the NiV G tetramer remains similar following ephrinB2 binding, SHG and HDX-MS are likely monitoring changes in G conformation dynamics or more subtle conformational changes at specific residues rather than large-scale conformational changes that would move the RBDs away from the stalk domain. Notably, movement of the lower two globular domains into a "heads-up" conformation, previously proposed as a mechanism for paramyxovirus activation, was not observed. We note that crystal structures of receptor-bound G do not reveal significant conformational changes in the G RBD[8, 15], indicating that the allosteric effects on peptide exchange may depend on the presence of the stalk domain and/or be due to dynamic differences in these structural regions rather than net conformational changes. The discovery of multiple protein-ligand interactions that cause allosteric effects at distal sites in the protein without apparent conformational change supports the line of thought that allostery can be propagated solely via changes in the thermal fluctuations of the protein[39, 40]. It may be that this is the case for NiV G, at least during the initial stage of activation when only an ephrinB2 monomer binds. Whether large-scale conformational changes are induced by dimeric ephrinB2-Fc binding in G was not possible to address in this study given the difficulty in imaging aggregated complexes (Supplementary Fig. 6b).

Negative stain EM revealed that NiV G is an asymmetric tetramer with an apparent dimer of globular domains at the far end of the stalk and a separated pair of globular domains in the middle of the stalk, which differs significantly from dimer-of-dimer structures observed in the related PIV5 and NDV HN proteins[22, 23]. While the observed HDX-MS data suggest a potential route for the propagation of dynamic changes from the receptor binding site to the stalk, through a lateral face of the RBD, the asymmetric structure of the G tetramer complicates this interpretation. Exchange rates from some of the NiV G peptides may represent a mixture of at least 2 different populations based on distinct interactions within the G tetramer. Nonetheless, the propagation of allosteric changes within both the RBD and stalk domains provides a compelling explanation for G activation and enables further studies of the functional roles of the G residues involved.

Molecular dynamics simulations of binding of an ephrinB2 monomer to NiV G identified residues that could make up a potential signaling pathway within the RBD and also resulted in little overall change in NiV G conformation[41, 42]. These computational studies identified residues 203–211 as undergoing the most significant net conformational changes upon ephrinB2 binding. This region is part of a peptide that undergoes a mild decrease in H/D exchange upon ephrinB2 binding (Fig. 5a, b and Supplementary Fig. 8a). Molecular dynamics simulations of

ephrinB2 binding to the crystallographic NiV G dimer resulted in significant reorientation of the dimer[43]. It is interesting to note the correlation in solvent accessibility of key peptides in these simulations with the H/D exchange rates in our study. NiV G residues unique to the ephrinB2-unbound crystallographic NiV G dimer interface are largely located in the two adjacent RBD peptides which show an increase in H/D exchange (Supplementary Fig. 8b), while residues unique to the ephrinB2-bound dimer interface following MD simulation include peptides that undergo decrease in H/D exchange upon receptor binding (Supplementary Fig. 8c).

The structure of the fully-activated NiV G-ephrinB2 signaling complex is still unknown, but what is known about ephrinB2 signaling via its native Eph receptors is highly suggestive of higher level oligomerization of ephrinB2. EphrinB2 is thought to form loosely associated dimers prior to stimulus, and during a signaling event forms 2:2 clusters with EphR, followed by higher order multimerization[9, 44]. While ephrinB2-expressing pseudovirions were able to cause NiV G and F displayed on pseudovirions or mammalian cells to undergo conformational changes and F to transition to its postfusion form, addition of soluble ephrinB2-Fc protein to was unable to trigger F, and induced less G conformational change[18]. Currently there is no direct observation of ephrinB2 clustering in response to NiV G binding or vice versa. However, clustering of HN and F from human parainfluenza virus 3 in response to cell surface sialic acid receptors has been observed in live cells[45], suggesting an analogous mechanism may occur in henipaviruses. These findings in total allow us to speculate on a sequence of NiV G-activating events from initial ephrinB2 monomer contact, binding of both heads of ephrinB2 dimers to NiV G globular domains, and finally events that can only occur with ephrinB2 clustering and/or presence of a target cell membrane.

## Methods

**Cloning of NiV G and ephrinB2 constructs**. A NiV G ectodomain construct consisting of the gp64 signal peptide, His6-tag, enterokinase cleavage site, and NiV G residues 71–602 was synthesized (Supplementary Table 2) and cloned into the SalI and NotI restriction sites of pENTR1A, and transferred to expression vector pcdnaDEST40 by Gateway cloning with LRII Clonase (Life Technologies). Site-directed mutants of the NiV G ectodomain were created by Gibson assembly with the mutant codon in the overlapping ends of the PCR-generated Gibson assembly fragments. An ephrinB2-Fc construct (ephrinB2-Fc) consisting of residues 1–229 of human ephrinB2-Fc, a Factor XA cleavage site, and residues 100–330 of human IgG₁ was synthesized and cloned into the EcoRI and NheI sites of pTT5 (National Research Council of Canada). An ephrinB2 monomer construct, His6-ephrinB2-167, consisting of the gp64 signal peptide, His6-tag, enterokinase cleavage site, and ephrinB2 residues 25–167 was synthesized and cloned into the EcoRI and NotI sites of pTT5 (National Research Council of Canada). The His6-tag was replaced with a S-tag using a Q5 site-directed mutagenesis kit (NEB) to generate the Stag-ephrinB2-167 construct. The entire plasmid was amplified by PCR with the S-tag encoding bases split between two back-to-back primers, followed by blunt-end ligation. An ephrinB2 monomer construct (ephrinB2-229) consisting of the gp64 signal peptide, S-tag, enterokinase cleavage site, and ephrinB2 residues 25–229 was created by PCR overlap extension. A PCR fragment containing ephrinB2-167 amplified from the Stag-ephrinB2-167-pTT5 plasmid and a PCR fragment containing an overlap with the 3′ end of fragment ephrinB2-167 followed by ephrinB2 residues 168–229 amplified from ephrinB2-Fc-pTT5 plasmid was combined in a PCR reaction to generate a single fragment. The fragment was digested with EcoRI and NotI and cloned into the same restriction sites in pTT5.

**Expression and purification of NiV G ephrinB2 constructs**. HEK 293F cells grown in Freestyle media (Life Technologies) were transiently transfected with NiV G pcdnaDEST40 plasmid at high cell density according to a protocol described in ref. [46]. Cell culture medium was harvested 5 days after transfection and dialyzed with 25 mM sodium phosphate pH 7.6, 200 mM NaCl, 10 mM imidazole. NiV G was purified from the medium by Ni-NTA chromatography and eluted with a stepwise imidazole gradient. The peak fractions were further purified by size exclusion on a Superdex S200 column with 25 mM Tris pH 8.0, 200 mM NaCl, 100 mM imidazole running buffer.

HEK 293 6E cells (National Research Council of Canada) grown in Freestyle media containing 0.1% Kollifor-188 (Sigma) and 25 μg mL⁻¹ G418 (Invivogen)

were transiently transfected with the ephrinB2-pTT5 plasmid according to the manufacturer's instructions (National Research Council of Canada). Cell culture medium was harvested 5 days after transfection and dialyzed with 25 mM Tris, pH 7.5, 200 mM NaCl. For ephrinB2-Fc, the dialyzed medium was loaded onto Pierce Protein A Plus beads (Thermo Scientific) and eluted with 3.5 M magnesium sulfate, 25 mM MES pH 6.6, followed by addition of 1 M Tris pH 8.0 to a final concentration of 100 mM. For ephrinB2-167 and ephrinB2-299, the dialyzed medium was loaded onto Protein S beads (EMD Millipore) and eluted with 3.0 M magnesium sulfate, 100 mM Tris pH 7.5. The eluates were concentrated and buffer exchanged into Tris-buffered saline (TBS, 50 mM Tris pH 7.5, 150 mM sodium chloride), and further purified by size exclusion chromatography on a 10/300 GL S200 column (GE Healthcare Life Sciences) with TBS running buffer.

**Labeling of NiV G with SHG-active dye**. NiV G was lysine-labeled with SHG-active dye (Biodesy, Inc.) via succinimidyl ester chemistry. NiV G was buffer exchanged into 100 mM sodium bicarbonate, pH 7.5, 150 mM sodium chloride prior to labeling. The ratio of dye to protein was 2X, resulting in a similar degree of labeling between wildtype and the mutant proteins of ~ 1.7 dyes/monomer. Unbound dye was removed and NiV G was exchanged into TBS using 0.5 mL ZebaSpin 7 K MWCO desalting columns (Thermo Scientific). The sites and degree of labeling of NiV G residues were determined by Martin-Protean (Princeton, New Jersey, USA) using LC-MS of denatured, reduced, and deglycosylated NiV G digested with trypsin and chymotrypsin.

**SHG measurements**. Supported lipid bilayers were formed by fusion of Ni-NTA lipid-containing small unilamellar vesicles in TBS to the well surface of 384-well Biodesy read plates. SHG-labeled-NiV G was bound to the lipid bilayer in 20 μL of TBS at a concentration of 0.5 μM overnight at 4 °C, followed by washing with TBS. SHG measurements were made on a Biodesy Delta instrument. p-polarized laser light is directed through a prism array to create an evanescent wave by total internal reflection at the well bottom surface. SHG-labeled-NiV G was illuminated by the evanescent wave, and the resulting signal from the label was detected by a photomultiplier tube. EphrinB2 protein (ephrinB2-Fc, His6-ephrinB2-167 with the His6-tag removed by enterokinase cleavage and Ni-NTA cleanup, or ephrinB2-229 (10881-HCCH, SinoBiological Inc.) was added from a reagent plate and mixed by the instrument's automated liquid handler. The change in SHG was monitored over regular time intervals. ΔSHG in % change was calculated using $(I_{final} - I_{initial})/I_{initial} \times 100$.

**ELISA measurements of ephrinB2 binding to NiV G**. NiV G ectodomain constructs were bound via their N-terminal His6-tags to Ni-NTA functionalized ELISA plates at 0.3 μg mL⁻¹ (Thermo Scientific) and titrated with ephrinB2 constructs. EphrinB2-Fc binding was detected by 1:3000 dilution anti-human IgG-HRP antibody conjugate (GtxHu-004-EHRPX, ImmunoReagents Inc.), and S-tag ephrinB2-167 and ephrinB2-229 were detected by 1:4000 diluted anti-S tag antibody (#12774, Cell Signalling Technologies) followed by 1:10,000 diluted anti-rabbit IgG-HRP antibody conjugate (AP187P, EMD Millipore). All ELISA incubations were performed in 100 μL volumes in buffer consisting of 1% BSA, 1X PBS (Corning Cellgro) and 0.01% Tween-20 for 1 h, followed by 3 × 250 μL washes of 1X PBS, 0.01% Tween-20. Color development at 630 nM was monitored using a Synergy 4 platereader (Biotek) following addition of HRP substrate (KPL). Curves were fit with Prism (Graphpad Software Inc.) according to a one site-specific binding model.

To measure the competitive binding of wildtype and C387A NiV G to ephrinB2-Fc, wildtype NiV G ectodomain was adsorbed to polystyrene ELISA plates in 10 mM Tris pH 8.5 at 1 μg mL⁻¹, then the surface was blocked with 1.5% BSA in TBS. Mixtures with increasing concentration of competitor NiV G with 0.080 nM of ephrinB2-Fc were added, and ephrinB2-Fc binding to NiV G on the plate was detected as above. Curves were fit with Prism (Graphpad Software Inc.) to a one site-fit $K_i$ model.

**ELISA of conformation-specific antibody binding to NiV G**. NiV G ectodomain constructs were bound at 0.3 μg mL⁻¹ via their membrane-proximal His6-tags to Ni-NTA functionalized ELISA plates (Thermo Scientific) and titrated with ephrinB2 constructs. mAb213 or mAb45 was bound at 1:6000 dilution followed by incubation with anti-rabbit IgG-HRP antibody conjugate (AP187P, EMD Millipore) at 1:10,000 dilution. The HRP signal was measured as described above. mAb45 binding curves were fit with Prism (Graphpad Software Inc.) to a log (agonist) vs. response-variable slope model with constraint of the bottom value to the lowest absorbance value. mAb213 binding curves were fit with Prism (Graphpad Software Inc.) to a log(inhibitor) vs. response-variable slope model with no constraints.

**Negative stain electron microscopy of NiV G mutants**. All samples were diluted in 25 mM Tris, pH 8.0, 200 mM NaCl, 100 mM imidazole to 0.01 mg mL⁻¹ and adsorbed on glow-discharged, carbon-coated copper grids. The samples were then stained with 2% uranyl formate. Electron micrographs were obtained on a JEOL 1400 electron microscope at 120 keV and recorded on a GATAN Ultrascan 4000 CCD camera with no binning.

**EM 2D class averages of NiV G and NiV G-ephrinB2 complexes**. 2.5 µL of 0.01 mg mL$^{-1}$ apo-NiV G in TBS was loaded onto glow-discharged, carbon-coated copper grids and stained with 2% uranyl acetate. His6-ephrinB2-167 was mixed with NiV G at a 5:1 molar ratio at a NiV G protein concentration of 0.12 mg mL$^{-1}$, and ephrinB2-Fc was mixed with NiV G at a 10:1 molar ratio at a NiV G protein concentration of 0.9 mg mL$^{-1}$ prior to concentration with 10 K MWCO Corning Spin-X concentrators to a 100 µL volume for loading onto a GL5/150 S200 column with 3 mL bed volume. The complex peak of His6-ephrinB2-167-NiV G was concentrated to 0.08 mg mL$^{-1}$ for storage. The complex peak of ephrinB2-Fc-NiV G was determined to be 0.09 mg mL$^{-1}$ and stored without concentration. NiV G-ephrinB2 complexes were diluted to 0.01 mg mL$^{-1}$ for adsorption onto glow-discharged carbon-coated copper grids. 140 electron micrographs were obtained on a Tecnai TF20 electron microscope at 200 keV. Ctffind 4[47] was used to determine the defocus value of each micrograph, ranging from 27,393 Å to 49,501 Å. 55,596 particles were picked without reference using DoGPicker[48]. Particle images were windowed out in dimensions of $128 \times 128$ square pixels (pixel size of 4.28 Å), followed by 2 rounds of 2-dimensional class averaging with Relion 1.4[49]. 23,025 good particles were kept in the final round of classification. The total number of particles picked for the His6-ephrinB2-167-NiV G complex was 47,666, yielding 40,015 good particles kept in the final round of classification. The location of the α-helical stalk portions of NiV G were estimated by calculating the length of an α-helix with the number of residues of the helical portion of the NiV G stalk (71–138, corresponding to 102 Å), and placing an appropriately-scaled graphic on a representative 2D class average of the NiV G tetramer based on the pixel size of 4.4 Å.

**Hydrogen-deuterium exchange mass spectrometry**. 12 µM NiV G ectodomain was used to determine sequences of peptides obtained in HDX-MS after pepsin digestion. 5 µM NiV G ectodomain was used as the apo-NiV G HDX-MS sample, and 5 µM NiV G ectodomain mixed with 2x molar excess of His6-ephrinB2-167 was used as the ephrinB2-bound NiV G HDX-MS sample. All protein samples were prepared in deuterated TBS, pD 7.5. Deuterated TBS was prepared by lyophilization of TBS and resuspension in D$_2$O (Cambridge Isotopes). Triplicate 5 µL protein samples were mixed with 55 µL deuterated TBS at 25 °C, quenched after 0.5, 1, 2, and 5 min with 125 mM sodium phosphate monobasic, pH 2.6, 250 mM TCEP for 1 min, and injected into a Waters G2-Si HDX-MS system (Waters, Corp.) by a LEAP H/DX PAL autosampler. Protein samples were digested on-column with immobilized pepsin (Pierce) at 0 °C, then separated by liquid chromatography on a Waters NanoACQUITY UPLC BEH C18 column with a 7–85% acetonitrile gradient in 0.1% formic acid. Peptides were analyzed by electrospray ionization mass spectrometry with ion mobility separation in a Synapt G2-Si quadrupole time-of-flight mass spectrometer. Peptides were identified with ProteinLynx Global Server (Waters Corp.). Mass spectra were assigned and H/D exchange was determined with DynamX 3.0 (Waters Corp.). The back exchange was < 30% and the data were corrected for this loss.

**Data availability**. All relevant data are available from the corresponding author upon reasonable request.

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

## Acknowledgements

We thank Hector Aguilar and Benhur Lee for providing the mAb213 and mAb45 antibodies. We thank Bason Clancy for preparation of Fig. 1b, c. This work was supported by NIH grants GM61050 (T.S.J.), GM07194 (Z.H.Z.), S10D012966 (E.A.K.) and the Howard Hughes Medical Institute (R.A.L.).

## Author contributions

J.J.W.W., J.S. and T.S.J. conceived the project. J.J.W.W. designed, performed, and analyzed data from SHG and ELISA experiments, performed single-particle analysis of NiV G EM images, processed HDX-MS data, and carried out protein purification. T.A.Y. designed and performed SHG experiments and data analysis. J.Z. and S.L. collected EM datasets of apo- and ephrinB2-bound wildtype NiV G and assisted in data processing. G. P.L. collected EM images of NiV G mutants. E.A.K. performed HDX-MS experiments and processed HDX-MS data. J.J.W.W. and T.S.J. wrote the manuscript. T.A.Y., J.S., R.A.L., E.A.K., Z.H.Z. and T.S.J. contributed to experimental design and manuscript editing.

## Additional information

**Competing interests:** T.A.Y. is an employee of Biodesy, Inc. J.S. is the Founder and Chief Scientific Officer of Biodesy, Inc. The remaining authors declare no competing financial interests.

