## [Peer Review file · Nature Communications]

Editorial note: Figure on p. 9 is reprinted from *Curr. Op. Virol.* 5:24, Jardetzky and Lamb (2014), with permission from Elsevier.

Reviewers' comments:

Reviewer #1 (Remarks to the Author):

In this manuscript the authors use a novel spectroscopic technique coupled with classical antibody binding and hydrogen exchange experiments to detail conformational changes in the G protein of Nipah virus upon the binding of monomeric or dimeric receptor. While the data itself appears reasonably solid the data as presented is observational and doesn't lead (or perhaps even advance) a detailed mechanistic model.

Reviewer #2 (Remarks to the Author):

This paper seeks to understand allosteric action in Nipah virus attachment protein G caused by binding of monomeric ephrinB2.

As I understand it the logic goes as follows. SHG and EM indicate that there are conformational changes associated with monomeric ephrinB2 binding even though binding of 2 conformationally sensitive mAbs are not affected as with dimeric ephrinB2 binding. HDX-MS experiments indicate changes in conformation or H-bonded stability at regions remote from the binding site of ephrinB2 apparently not mediated by intervening conformational changes. Such a result is interesting in general even more so where it elucidates the mode of infection of an important class of viruses. I think this paper should be published but I think it could be improved as discussed below.

I found this paper difficult to read. Given the large number of techniques involved, a brief outline at the beginning of how the argument flows from the various results would be useful.

While I don't doubt the general conclusions that some residues remote from ephrinB2 binding lose some degree of exchange protection upon binding. I'm disappointed with the HDX-MS data. The temperature and pD of the exchange experiments are not given making it impossible to assess the degree of structural protection for the residues observed. Exchange was actually measured only for a small fraction of residues in each peptide the rest being too fast or too slow for the narrow time range observed. Interestingly, for most of the peptides shown with and without binding exchange curves are very nearly parallel with only a small vertical offset. This suggests that the change in exchange rates rather than being spread over the entire peptide may involve only one or two residues going from too fast to measure to too slow to measure or vice versa.

A minor comment, the black line and especially the red line in Fig 5 are very hard to see at printed size & will only be worse when reduced to fit a journal page.

Reviewer #3 (Remarks to the Author):

The ms by Wong et al., utilizes a relatively new technique, second harmonic generation (SHG), in combination with HDXMS, EM, and antigenicity profiling to describe the subtle conformational changes in the Nipah Virus attachment protein G, that initiates the viral fusion process by triggering the F fusion protein. Protein G binds to the host receptor ephrinB2, and the authors show that monomeric ephrinB2 induces subtle allosteric changes that propagate to the stalk in G. The conformational changes inferred from changes in antigenicity are consistent with previously published studies of full-length protein G on the surface of membranes. Overall the data are consistent and appear to elucidate previously hidden monomeric ephrinB2 receptor-mediated allosteric changes in protein G, primarily in the stalk region that tethers the protein G tetramer together.

While the SHG data is impressive and the negative stain EM images are revealing I find the HDXMS data to be the most informative, particularly changes in the stalk region of G. However, because there is only a crystal structure of the globular monomeric G receptor binding domain bound to ephrinB2 and no structural information about the stalk the data is presented in a vacuum

and can only speculate mechanism. Hence, this story would have been more compelling on a system with more complete structural definition. The study is not without merit and should be of narrow interest to the Nipah Virus research community.

Although subtle, this ms reads like an advertisement for the SHG system, and highlights results using this technique even though the HDX-MS data are more complete and compelling. Further, two of the authors have financial ties to Biodesy and declare potential conflicts of interest.

Reviewer #4 (Remarks to the Author):

The authors examined the conformational changes in NiV G protein induced by ephrinB2 binding using SHG, antibody binding, EM and HDX-MS, and suggested that dimeric and monomeric ephrinB2 activate NiV G differently. Conformational changes in NiV G triggered by ephrinB2 were observed using SHG and conformation-sensitive mAbs. On the other hand, monomeric ephrinB2 induced conformational changes in NiV G that were detected by SHG and HDX-MS, but not mAbs or EM. More interestingly, the authors revealed allosteric changes in the RBD and F-activating stalk domains of NiV G through receptor binding using H/D exchange. The study is well presented and the results in general justify the conclusions. The following are some minor concerns.

Fig. 1a presents only the globular domain of NiV G. It would be good to see the relevant positions of the mutants and the SHG labels to the stalk domain as well in such a figure (even just in a schematic way). It seems the full ectodomain of NiV G is used in this study. This may also help the readers to picture the long-range allosteric effects as well.

The authors mention B-class ephrins are thought to form loose dimers. Dimeric ephrinB2 is achieved in this study through Fc fusion. What interactions hold ephrinB2 dimers together in nature? Is ephrinB2 oligomerization concentration-dependent? In this study, is it possible the monomeric ephrinB2 forms oligomer at high concentration? If it does, would it affect the evaluation of the experimental results (e.g. when μM range of ephrinB2 is used)?

The NiV G mutants/labelings used in the study are all in the RBD region. Have the authors considered testing mutants/labelings in the stalk domain to monitor conformational changes in the stalk region and to back up the observations seen in H/D exchange experiments?

Do the authors have any thoughts on the similar antibody-sensitive conformational changes observation with full length G and ectodomain G? Does this suggest that the transmembrane region of G is not involved in receptor binding and allostery? Also, what is the oligomerization stage of the full length G in the previous studies?

The binding affinities of ephrinB2-167 and ephrinB2-229 are 11.9 nM and 33.2 nM (what are the error?), which presents a 3-fold difference. I feel this is not quite negligible, especially the shortened ephrinB2 is binding stronger. The authors may want to comment on this.

EphrinB2-167 instead of ephrinB2-229 is used for EM and H/D exchange. Any specific reason? Would the authors expect any differences in the results if ephrinB2-229 is used?

If F protein forms pre-complex with G, do the authors consider adding F protein in their experiments?

Do the authors have any thought/discussions on why dimeric ephrinB2 is necessary to induce the full conformational changes of G?

Reviewer #5 (Remarks to the Author):

Wong et al describe a multi-technological study with several accomplishments. 1. It supports the important conclusions that Liu et al drew and published in 2013 and 2015: that a series of receptor-induced conformational changes occur in NiV-G that result in membrane fusion. 2. It introduces a new technological advances to the field of virology. 3. It identifies new regions in NiV-G that change their orientation and/or conformation upon receptor binding. The manuscript is significant and well written. This study is particularly strong because it validates prior conclusions while introducing several new techniques. In my opinion, no new experiments are needed, since the main piece of data missing, the effects of dimeric ephrinB2 on NiV G conformational changes by EM, was at least tried, although it resulted in aggregates and no data. I have a few comments to further improve this manuscript.

Relatively major:

1. In nature, ephrinB2 is thought to be primarily dimeric. Therefore, the conclusions drawn from using monomeric ephrinB2 need to be taken with a grain of salt. It is interesting that monomeric ephrinB2 can induce some conformational changes. However, it appears that the authors believe that the interactions of NiV-G with ephrinB2 are sequential: first with monomeric, then with dimeric, and then with larger-oligomeric ephrinB2, and although this is quite logical, there is no direct evidence that shows that this really happens in nature, given that ephrinB2 is believed to be dimeric. Although this proposed mechanism is interesting, I would make sure that it is clear that it is speculative.

2. Towards the bottom of page 7, second to last paragraph, the authors say: "LC-MS analysis of the labeled proteins indicated that the wild-type and mutant proteins were labeled in a similar manner across all constructs." However, there were definitive differences in labeling of K386 for mutants K376A, C387A, and Q388A. This makes sense since these mutations are in the relative vicinity of K386. I think this should be acknowledged upfront.

3. In the last paragraph of page 8, it should be mentioned that the lower level of ephrinB2 binding to mutant C387A observed in this study is consistent with the lower level of binding of ephrinB2 to mutant C387A published in Liu et al, 2013 (supplementary Fig. 4). Also "lack of ephrinB2 binding" is probably too strong of a way to describe the lower binding levels of ephrinB2 to C387A.

4. In the middle of the last paragraph of page 15 it is said that "detachment of the lower two globular domains from the stalk into a heads up conformation previously proposed as a mechanism for paramyxovirus activation, is absent." What do the authors mean by "is absent"? Are they referring to the lack of changes observed by their EM studies using soluble NiV-G and monomeric ephrinB2? If this is the case, this sentence should be made clearer. Also, it should be acknowledged that the lack of movements observed by EM is not so surprising, as EM is has very low sensitivity, and the proteins being used are soluble (G) and monomeric (ephrinB2), which are not their most natural states.

5. In general, more references should be introduced into the Discussion section. Very few are currently there. For example, include Landowski et al, 2014, which showed that receptor-induced conformational changes in NiV G occur not only on cell surfaces, as in Liu et al, 2013 and 2015, but also in virions. Other references should be added where appropriate.

Relatively minor:

1. At what concentrations was ephrinA1-Fc used in Figure 1f?

2. There are several examples where subfigures are mis-quoted or not quoted at all. For example,

Fig. 1f is never quoted, and its contents are sometimes referred to as Fig. 1d or 1e. Examples: Last paragraph in page 7, last paragraph in page 6, top paragraph in page 10, legend to Fig. 4, ... etc.

3. EphRB2 is a fairly unusual term. It is most common to use Eph to refer to the receptors for ephrins. The authors should consider the most standard nomenclature.

DETAILED RESPONSES TO REVIEWERS

Reviewer #1:

In this manuscript the authors use a novel spectroscopic technique coupled with classical antibody binding and hydrogen exchange experiments to detail conformational changes in the G protein of Nipah virus upon the binding of monomeric or dimeric receptor. While the data itself appears reasonably solid the data as presented is observational and doesn't lead (or perhaps even advance) a detailed mechanistic model.

We believe that our data shows significant mechanistic insights into the NiV G response to ephrinB2 binding that go well beyond previous studies. Our SHG and HDX-MS data reveal changes in G protein dynamics and conformation in response to ephrinB2 binding. In the case of HDX-MS, the data allow us to identify specific peptide regions involved, identifying allosteric changes and indicating how their dynamics are affected by receptor binding. Our electron microscopy 2D class averages reveal a G tetramer structure that is asymmetric and different from other paramyxovirus attachment protein tetramers. Our EM data also show that monomeric ephrinB2 is insufficient to cause large-scale conformational changes in G. Together with our NiV G conformation-sensitive antibody ELISA assays and SHG assays showing differences between monomeric and NiV G, the totality of the data allow us to propose a mechanistic model where dimers and higher order oligomers may be necessary for the large scale conformational changes that reveal the stalk region and activate F.

Reviewer #2:

This paper seeks to understand allosteric action in Nipah virus attachment protein G caused by binding of monomeric ephrinB2.

As I understand it the logic goes as follows. SHG and EM indicate that there are conformational changes associated with monomeric ephrinB2 binding even though binding of 2 conformationally sensitive mAbs are not affected as with dimeric ephrinB2 binding. HDX-MS experiments indicate changes in conformation or H-bonded stability at regions remote from the binding site of ephrinB2 apparently not mediated by intervening conformational changes. Such a result is interesting in general even more so where it elucidates the mode of infection of an important class of viruses. I think this paper should be published but I think it could be improved as discussed below.

I found this paper difficult to read. Given the large number of techniques involved, a brief outline at the beginning of how the argument flows from the various results would be useful.

We have modified the final summary paragraph of the introduction to better describe the flow, rationale and connections between the main findings from the different techniques used.

While I don't doubt the general conclusions that some residues remote from ephrinB2 binding lose some degree of exchange protection upon binding. I'm disappointed with the HDX-MS data. The temperature and pD of the exchange experiments are not given making it impossible to assess the degree of structural protection for the residues observed. Exchange was actually measured only for a small fraction of residues in each peptide the rest being too fast or too slow for the narrow time range observed. Interestingly, for most of the peptides shown with and without binding exchange curves are very nearly parallel with only a small vertical offset. This suggests that the change in exchange rates rather than being spread over the entire peptide may involve only one or two residues going from too fast to measure to too slow to measure or vice versa.

The temperature and pD of the exchange experiment has now been added to the methods. The pD is the same as the pH of the protonated buffer, as the buffer was prepared by lyophilization and resuspension in D₂O. The reason for the short timepoints is because we are sampling the surface of the protein and the dynamic parts of the structure. This time window is the most informative for the timescale of exchange of these regions of the protein that change upon protein-protein interactions. It is correct that it is very likely that only a few amides per peptide are affected, which is expected

and what was found in (Ramsay, et al, 2017. J Mol Biol. 2017. 429:999-1008. doi: 10.1016/j.jmb.2017.02.017).

A minor comment, the black line and especially the red line in Fig 5 are very hard to see at printed size & will only be worse when reduced to fit a journal page.

Fig. 5d has been enlarged to allow easier viewing.

Reviewer #3:

The ms by Wong et al., utilizes a relatively new technique, second harmonic generation (SHG), in combination with HDXMS, EM, and antigenicity profiling to describe the subtle conformational changes in the Nipah Virus attachment protein G, that initiates the viral fusion process by triggering the F fusion protein. Protein G binds to the host receptor ephrinB2, and the authors show that monomeric ephrinB2 induces subtle allosteric changes that propagate to the stalk in G. The conformational changes inferred from changes in antigenicity are consistent with previously published studies of full-length protein G on the surface of membranes. Overall the data are consistent and appear to elucidate previously hidden monomeric ephrinB2 receptor-mediated allosteric changes in protein G, primarily in the stalk region that tethers the protein G tetramer together.

While the SHG data is impressive and the negative stain EM images are revealing I find the HDXMS data to be the most informative, particularly changes in the stalk region of G. However, because there is only a crystal structure of the globular monomeric G receptor binding domain bound to ephrinB2 and no structural information about the stalk the data is presented in a vacuum and can only speculate mechanism. Hence, this story would have been more compelling on a system with more complete structural definition. The study is not without merit and should be of narrow interest to the Nipah Virus research community.

We agree that the HDX-MS provides the most detailed insights into the effects of receptor binding on G, although the interpretation of these results also relies on our other comprehensive observations.

It is important to note that the HDX-MS experiments conducted here would be very difficult to perform with HN proteins, for which more complete structures are available. HN interactions with receptors (sialic acid) are many orders of magnitude weaker (mM) than the G:EphB2 interaction. Because of the weaker Kd, it would require impractically high concentrations of ligand (e.g. sialyl-lactose) to have a chance of impacting HN HD exchange rates.

Based on our electron microscopy class averages and sequence alignment with attachment proteins from other members of the paramyxovirus family with available crystal structures, we believe that it is reasonably certain that the lower portion of the stalk forms a bundle of 4-alpha helices similar that of known crystal structures. The EM class averages show a linear stalk with a fair amount of electron density, indicating some secondary structure. Sequence alignment of NiV G with attachment proteins from other members of the paramyxovirus family (including PIV5 and NDV HN, where crystal structures of the ectodomain including the stalk are available) shows that NiV G also has heptad and 11-mer repeats of non-polar amino acids which form the core of the 4-helix bundle (Maar et al, 2012. J.Virol. 86:6632. Fig. 9B).

Although subtle, this ms reads like an advertisement for the SHG system, and highlights results using this technique even though the HDXMS data are more complete and compelling. Further, two of the authors have financial ties to Biodesy and declare potential conflicts of interest.

We have rewritten the introduction to better explain how the SHG results provided compelling data to undertake the HDX-MS studies. The conformation-sensitive antibody binding and the EM data indicated that there were no significant changes in the G structure upon monomeric ephrinB2 binding. Based on that data alone, we would have assumed that HDX-MS would only reveal the footprint of ephrinB2 binding on the G receptor binding site. In fact, the EM and antibody data strongly indicated that monomeric receptor was not inducing any conformational changes in G at

all. Because the SHG data clearly indicated that conformational changes were occurring, conducting the HDX-MS experiments became much more important to pursue. Scientifically, the SHG experiments were absolutely central to advancing our understanding. Because SHG is relatively novel and has distinct experimental advantages, we feel that is appropriate and helpful to include sufficient background and explanation to the readers.

Reviewer #4:

The authors examined the conformational changes in NiV G protein induced by ephrinB2 binding using SHG, antibody binding, EM and HDX-MS, and suggested that dimeric and monomeric ephrinB2 activate NiV G differently. Conformational changes in NiV G triggered by ephrinB2 were observed using SHG and conformation-sensitive mAbs. On the other hand, monomeric ephrinB2 induced conformational changes in NiV G that were detected by SHG and HDX-MS, but not mAbs or EM. More interestingly, the authors revealed allosteric changes in the RBD and F-activating stalk domains of NiV G through receptor binding using H/D exchange. The study is well presented and the results in general justify the conclusions. The following are some minor concerns.

Fig. 1a presents only the globular domain of NiV G. It would be good to see the relevant positions of the mutants and the SHG labels to the stalk domain as well in such a figure (even just in a schematic way). It seems the full ectodomain of NiV G is used in this study. This may also help the readers to picture the long-range allosteric effects as well.

Unfortunately, it is not possible to accurately position the mutants and SHG labels relative to the stalk because there is no crystal structure available of the NiV G ectodomain including the stalk, and the electron microscopy class averages are still too low resolution to orient the globular domains. The legend to Fig. 1d has been modified to specify the stalk and the globular domains of the NiV G schematic.

The authors mention B-class ephrins are thought to form loose dimers. Dimeric ephrinB2 is achieved in this study through Fc fusion. What interactions hold ephrinB2 dimers together in nature? Is ephrinB2 oligomerization concentration-dependent? In this study, is it possible the monomeric ephrinB2 forms oligomer at high concentration? If it does, would it affect the evaluation of the experimental results (e.g. when μM range of ephrinB2 is used)?

Membrane attachment is thought to be important for ephrinB2 dimerization and higher order oligomerization. We currently cannot find direct evidence for oligomerization of the monomeric ephrinB2 in solution elsewhere in the literature, though other ephrinB2 family members have been shown to be monomeric in solution, e.g. ephrinB1 and ephrinB3. Our size exclusion traces of both constructs of the ephrinB2 monomer used in this study (ephB2-167 and ephB2-229) shows that they are monomeric.

The NiV G mutants/labelings used in the study are all in the RBD region. Have the authors considered testing mutants/labelings in the stalk domain to monitor conformational changes in the stalk region and to back up the observations seen in H/D exchange experiments?

This is an excellent suggestion and we are in the process of conducting site-specific labeling studies along these lines. However, these experiments are beyond the scope of this current report.

Do the authors have any thoughts on the similar antibody-sensitive conformational changes observation with full length G and ectodomain G? Does this suggest that the transmembrane region of G is not involved in receptor binding and allostery? Also, what is the oligomerization stage of the full length G in the previous studies?

Our results indicate that the secreted G ectodomain and full length G protein undergo similar changes as detected by the conformational sensitive antibodies. However, we cannot discount the possibility that the transmembrane domain has some role in allostery that is not detected by these particular conformation sensitive antibodies, which recognize epitopes only in the head domain. Full-length wild-type NiV G has been shown in previous literature to be tetrameric, similar to the secreted ectodomain.

The binding affinities of ephrinB2-167 and ephrinB2-229 are 11.9 nM and 33.2 nM (what are the error?), which presents a 3-fold difference. I feel this is not quite negligible, especially the shortened ephrinB2 is binding stronger. The authors may want to comment on this.

The standard error of the calculated Kd values are 1.1 nM for ephrinB2-167 and 2.8 nM for ephrinB2-229 - this information is included in Supplementary Figure 5. It is difficult to ascribe biological significance to these observations, as neither construct shows any difference in oligomeric state on size exclusion columns, and the affinity difference is not large. Some of this difference could be due to protein quantitation using the BCA assay, which could show some variability based on amino acid composition. We did not feel that there was sufficient biological justification for investigating this relatively small affinity difference in greater detail.

EphrinB2-167 instead of ephrinB2-229 is used for EM and H/D exchange. Any specific reason? Would the authors expect any differences in the results if ephrinB2-229 is used?

We used the shorter ephrinB2 protein because this represents the minimal construct with high affinity interactions, potentially yielding fewer peptides that could overlap and obscure our analysis of the G protein. This clarification is included in the main text of the revised manuscript.

If F protein forms pre-complex with G, do the authors consider adding F protein in their experiments?

These are experiments that we have initiated and that are ongoing.

Do the authors have any thought/discussions on why dimeric ephrinB2 is necessary to induce the full conformational changes of G?

It may be that higher order oligomers caused by crosslinking of NiV G is necessary for full activation.

Reviewer #5:

Wong et al describe a multi-technological study with several accomplishments. 1. It supports the important conclusions that Liu et al drew and published in 2013 and 2015: that a series of receptor-induced conformational changes occur in NiV-G that result in membrane fusion. 2. It introduces a new technological advances to the field of virology. 3. It identifies new regions in NiV-G that change their orientation and/or conformation upon receptor binding. The manuscript is significant and well written. This study is particularly strong because it validates prior conclusions while introducing several new techniques. In my opinion, no new experiments are needed, since the main piece of data missing, the effects of dimeric ephrinB2 on NiV G conformational changes by EM, was at least tried, although it resulted in aggregates and no data. I have a few comments to further improve this manuscript.

Relatively major:

1. In nature, ephrinB2 is thought to be primarily dimeric. Therefore, the conclusions drawn from using monomeric ephrinB2 need to be taken with a grain of salt. It is interesting that monomeric ephrinB2 can induce some conformational changes. However, it appears that the authors believe that the interactions of NiV-G with ephrinB2 are sequential: first with monomeric, then with dimeric, and then with larger-oligomeric

ephrinB2, and although this is quite logical, there is no direct evidence that shows that this really happens in nature, given that *ephrinB2* is believed to be dimeric. Although this proposed mechanism is interesting, I would make sure that it is clear that it is speculative.

We don't think that the effects of monomeric ephrinB2 that we present are incompatible with the natural, dimeric forms of ephrinB2. Our interpretation of the data is that the individual receptor binding interactions with G subunits induces allosteric changes that precede and likely enable larger conformational changes in the tetramer. Even if the ephrinB2 encountered in nature is dimeric, each of the NiV G heads will have to engage one of the ephrinB2 heads, likely in some stepwise manner rather than simultaneously. The higher order oligomerization of ephrinB2 and NiV G is speculative at present, and the end of discussion has been modified to reflect that. In addition, a reference showing clustering of the fusion and attachment protein of a related paramyxovirus on live cells upon allowing receptor binding is added.

2. Towards the bottom of page 7, second to last paragraph, the authors say: "LC-MS analysis of the labeled proteins indicated that the wild-type and mutant proteins were labeled in a similar manner across all constructs." However, there were definitive differences in labeling of K386 for mutants K376A, C387A, and Q388A. This makes sense since these mutations are in the relative vicinity of K386. I think this should be acknowledged upfront.

The text has been changed to reflect the differences in labeling.

3. In the last paragraph of page 8, it should be mentioned that the lower level of ephrinB2 binding to mutant C387A observed in this study is consistent with the lower level of binding of ephrinB2 to mutant C387A published in Liu et al, 2013 (supplementary Fig. 4). Also "lack of ephrinB2 binding" is probably too strong of a way to describe the lower binding levels of ephrinB2 to C387A.

The reference to Liu et al, 2013 has been added. The text has been changed from "lack of ephrinB2 binding" to "weaker binding affinity"

4. In the middle of the last paragraph of page 15 it is said that "detachment of the lower two globular domains from the stalk into a heads up conformation previously proposed as a mechanism for paramyxovirus activation, is absent." What do the authors mean by "is absent"? Are they referring to the lack of changes observed by their EM studies using soluble NiV-G and monomeric ephrinB2? If this is the case, this sentence should be made clearer. Also, it should be acknowledged that the lack of movements observed by EM is not so surprising, as EM has very low sensitivity, and the proteins being used are soluble (G) and monomeric (ephrinB2), which are not their most natural states.

We have revised the text to make this point more clearly. We were referring to the lack of obvious conformational change in the NiV G tetramer observed by EM. The changes in previously proposed models (from a "heads down" to "heads up" position are quite dramatic, as illustrated below from Jardetzky and Lamb, 2014. Curr. Op. Virol. 5:24), and we believe a change of that magnitude should be observed by low resolution negative stain EM.

5. In general, more references should be introduced into the Discussion section. Very few are currently there. For example, include Landowski et al, 2014, which showed that receptor-induced conformational changes in NiV G occur not only on cell surfaces, as in Liu et al, 2013 and 2015, but also in virions. Other references should be added where appropriate.

Landowski et al, 2014 has been added to the Introduction section as comparison of soluble vs membrane attached ephrinB2 was not done in this study, and so was not as relevant to the main points of the final paragraph.

Relatively minor:

1. *At what concentrations was ephrinA1-Fc used in Figure 1f?*

0.6 μ M – this is indicated in the figure.

2. *There are several examples where subfigures are mis-quoted or not quoted at all. For example, Fig. 1f is never quoted, and its contents are sometimes referred to as Fig. 1d or 1e. Examples: Last paragraph in page 7, last paragraph in page 6, top paragraph in page 10, legend to Fig. 4, ... etc.*

References to Fig. 1f were added correctly. The Fig. 4 legend is on the next page after the figure.

3. *EphRB2 is a fairly unusual term. It is most common to use Eph to refer to the receptors for ephrins. The authors should consider the most standard nomenclature.*

EphRB2 in Fig. 1 has been changed to EphB2, and labels for ephrinB2 in figures have been changed to be more consistent.